

# 1 Variation and Trend of Nitrate radical reactivity towards

# 2 volatile organic compounds in Beijing, China

Hejun Hu[1], Haichao Wang[1, 2 *], Keding Lu[3*], Jie Wang[1], Zelong Zheng[1], Xuezhen Xu[1], Tianyu Zhai[3],
Xiaorui Chen[4], Xiao Lu[1, 2], Momei Qin[5], Xin Li[3], Limin Zeng[3], Min Hu[3], Yuanhang Zhang[3]
[1]School of Atmospheric Sciences, Sun Yat-sen University, and Southern Marine Science and
Engineering Guangdong Laboratory (Zhuhai), Zhuhai, 519082, China
[2]Guangdong Provincial Observation and Research Station for Climate Environment and Air Quality
Change in the Pearl River Estuary, Key Laboratory of Tropical Atmosphere-Ocean System (Sun
Yat-sen University), Ministry of Education, Zhuhai, 519082, China
[3]State Key Joint Laboratory of Environmental Simulation and Pollution Control, The State
Environmental Protection Key Laboratory of Atmospheric Ozone Pollution Control, College of
Environmental Sciences and Engineering, Peking University, Beijing, 100871, China.
[4]Department of Civil and Environmental Engineering, The Hong Kong Polytechnic University, Hong
Kong, China
[5]Jiangsu Key Laboratory of Atmospheric Environment Monitoring and Pollution Control,
Collaborative Innovation Center of Atmospheric Environment and Equipment Technology, Nanjing
University of Information Science and Technology, Nanjing, China
*Correspondence to:* Haichao Wang (wanghch27@mail.sysu.edu.cn), Keding Lu (k.lu@pku.edu.cn)

**ABSTRACT.** Nitrate radical ($NO_3$) is an important nocturnal atmospheric oxidant in the troposphere,
which significantly affects the lifetime of pollutants emitted by anthropogenic and biological
activities, especially volatile organic compounds (VOC). Here, we used one-year VOC observation
data obtained in urban Beijing in 2019 to look insight to the level, compositions and seasonal
variation of $NO_3$ reactivity $(k_{NO3})$. We show the hourly $k_{NO3}$ towards measured VOC highly varied
from $<10^{-4}$ to 0.083 s$^{-1}$ with campaign-averaged value (± standard deviation) of 0.0032 ± 0.0042 s$^{-1}$.
There was large seasonal difference in $NO_3$ reactivity towards VOC with the average of 0.0024 ±
0.0026 s$^{-1}$, 0.0067 ± 0.0066 s$^{-1}$, 0.0042 ± 0.0037 s$^{-1}$, 0.0027 ± 0.0028 s$^{-1}$ from spring to winter.
Alkenes such as isoprene and styrene accounted for the majority. Isoprene was the dominant species
in spring, summer, and autumn, accounting for 40.0%, 77.2% and 43.2%, respectively. Styrene only
played a leading role in winter with the percentage of 39.8%. Sensitivity study shows monoterpenes,
the species we did not measure, may account a large fraction of $k_{NO3}$. Based on the correlation
between the calculated $k_{NO3}$ and VOC concentrations in 2019, we established localized
parameterization schemes for predicting the reactivity by only using a part of VOC species. The
historical published VOC data was collected to reconstruct the long-term $NO_3$ reactivity in Beijing
by the parameterization method. The downward trend of $k_{NO3}$ during 2011-2020 may be responded to
the reduction of anthropogenic VOC emission. At last, we revealed that $NO_3$ dominated the nocturnal
VOC oxidation with 83% on the annual average in Beijing in 2019, which varied seasonally and was
strongly regulated by the level of $k_{NO3}$, nitrogen oxide and ozone. Our results improve the



understanding of nocturnal atmospheric oxidation in urban regions, and gain the knowledge of nocturnal VOC oxidation and secondary organic pollution.

## 1.  Introduction

Nitrate radical ($NO_3$) is the main nocturnal tropospheric oxidant (Brown and Stutz, 2012; Wayne et al., 1991), which is mainly formed in the reaction of $NO_2$ and $O_3$. During the daytime, a large amount of NO emitted by cities is oxidized into $NO_2$ by ozone and released into the atmosphere (R1), and $NO_2$ continues to be oxidized into $NO_3$ by $O_3$ (R2). $NO_3$ only presents a high concentration level at night because it has a rapid photolysis rate during the daytime (Stark et al., 2007). $NO_3$ can oxidize NO into $NO_2$ (R3). During the nighttime, $NO_3$ and $NO_2$ react to form nitrous pentoxide ($N_2O_5$) (R4), and $N_2O_5$ can be decomposed to $NO_3$ and $NO_2$ (R5), establishing a temperature-dependent equilibrium.

$$NO+O_3 \rightarrow NO_2+O_2 \qquad (R1)$$

$$NO_2+O_3 \rightarrow NO_3+O_2 \qquad (R2)$$

$$NO_3+NO \rightarrow 2NO_2 \qquad (R3)$$

$$NO_2+NO_3+M \rightarrow N_2O_5+M \qquad (R4)$$

$$N_2O_5+M \rightarrow NO_2+NO_3+M \qquad (R5)$$

The main removal of $NO_3$ from the gas phase is the reaction with NO (R3), solar photolysis (R6, R7), and volatile organic compounds oxidation (R8), forming complex products. In addition, $NO_3$ can be transformed into $N_2O_5$ and removed by heterogeneous hydrolysis (R9), providing an effective way to remove $NO_X$ and produce nitrate aerosol and nitryl chloride (Brown et al., 2004; Dentener and Crutz, 1993; Osthoff et al., 2008). The competition between R8 and R9 determines the fate of nocturnal nitrogen oxidation chemistry, which leads to the formation of different type secondary pollutants (Bertram and Thornton, 2009; Brown et al., 2006). Specifically, the degradation of VOC by $NO_3$, especially biogenic VOC (Ng et al., 2017), has been proven to be related to the formation of organic nitrate and secondary organic aerosols (SOA) (Goldstein and Galbally, 2009; Kiendler-Scharr et al., 2016).

$$NO_3+hv \rightarrow NO+O_2 \qquad (R6)$$

$$NO_3+hv \rightarrow NO_2+O^1D \qquad (R7)$$

$$NO_3+VOC \rightarrow products \qquad (R8)$$

$$N_2O_5+H_2O_{(aq)}/Cl^- \rightarrow HNO_3 + \varphi ClNO_2 \qquad (R9)$$

The high $NO_3$ concentration and fast reaction rate make $NO_3$ responsible for the sink of many



unsaturated hydrocarbons at night (Edwards et al., 2017; Ng et al., 2017; Yang et al., 2020). The $NO_3$
reactivity ($k_{NO3}$) towards VOC can be calculated by Eq. 1.
$$k_{NO3} = \sum k_i \times [VOC_i] \qquad \text{Eq. 1}$$
where the $[VOC_i]$ is VOC concentrations and $k_i$ is the corresponding reaction rate coefficients. Table
S1 gives the reaction rate coefficients of $NO_3$ with VOC (Atkinson and Arey, 2003). The $NO_3$
reactivity towards different VOC varies greatly, which is affected by the abundance of species and
the reaction rate coefficients. $NO_3$ reactivity towards VOC is also affected by temperature. Since
temperature not only affects the reaction rate coefficients but also the VOC concentrations in the
atmosphere, especially for the emission of biogenic VOC like isoprene and monoterpenes (Wu et al.,
2020), causing the variations of VOC species which dominate $k_{NO3}$ towards VOC in different
seasons.
The VOC species which dominant the $NO_3$ reactivity vary greatly between different regions. In
forests and rural areas, such as Pabstchum outside Berlin, Germany, the lush forests emit a large
amount of monoterpenes and isoprene, accounting for the majority of $k_{NO3}$, which ranged from
0.0025 to 0.01 $s^{-1}$ (Asaf et al., 2009); In semi-arid urban areas such as Jerusalem, the emissions of
BVOC are less due to the sparser vegetation, and the maximum of $NO_3$ reactivity was about 0.01 $s^{-1}$.
Phenol, cresol and some monoolefins emitted by road traffic are the main contributors (Asaf et al.,
2009). In the urban regions like Houston, the industrial emissions including isoprene and other
alkenes dominated the $NO_3$ reactivity (Stutz et al., 2010). In the suburbs of the city, the $k_{NO3}$ may be
jointly affected by anthropogenic and biological volatile organic compounds. For example, the $NO_3$
reactivity towards VOC in Xianghe, Beijing reached 0.024 ± 0.030 $s^{-1}$, with the maximum value of
0.3 $s^{-1}$ and minimum value of 0.0011 $s^{-1}$. Isoprene, styrene and 2-butene contributed to the majority
of the $k_{NO3}$ (Yang et al., 2020).
In addition to calculating $k_{NO3}$ by the measured VOC, an instrument was developed to directly
measure $k_{NO3}$ in the atmosphere (Liebmann et al., 2017). On this basis, they presented the first direct
measurement of $NO_3$ reactivity in the Finnish boreal forest in 2017 and concluded that the $NO_3$
reactivity was generally high with a maximum value of 0.94 $s^{-1}$, displaying a strong diel variation
with nighttime mean value of 0.11 $s^{-1}$ and daytime value of 0.04 $s^{-1}$ (Liebmann et al., 2018a). In 2018,
they presented the direct measurement in and above the boundary layer of a mountain site, with
daytime values of up to 0.3 $s^{-1}$ and nighttime values close to 0.005 $s^{-1}$ (Liebmann et al., 2018b). Most
importantly, the direct measurement revealed the existence of missing $NO_3$ reactivity in varies
regions, which indicated the missing $NO_3$ oxidation mechanisms, and largely improved the
understanding of nighttime chemistry.
Nevertheless, the field direct determination of $k_{NO3}$ is still extremely lacked, especially in urban
regions. Until now, most works about the VOC oxidation by $NO_3$ was usually based on short-term
investigations, and the analysis of nocturnal chemical process or reactivity was carried out based on
the data of a few weeks or several months. The studies of nighttime chemistry based on long-term
measurement data are scarce (Vrekoussis et al., 2007; Wang et al., 2023; Zhu et al., 2022). The
detailed VOC contributions to $k_{NO3}$, and the relationship between certain VOC and total $NO_3$
reactivity in a long-time scale are rarely studied. Our recent work reported that the increasing trend



of NO$_3$ production rate caused by the anthropogenic emission changes, while the long-term and
detailed NO$_3$ loss budget is still uncertain to some extent (Wang et al., 2023). Here, we attempt to
look insight to the level, variations and impacts of NO$_3$ reactivity by using the one-year measurement
of VOC in an urban site in Beijing, the role of unmeasured VOC species (monoterpenes) in the
contributions of NO$_3$ reactivity is also discussed. The long-term trend of NO$_3$ reactivity is estimated
by collecting the published VOC data and the proposed parameterization method. At last, the
regulation of NO$_3$ oxidation of nocturnal VOC in different seasons is further evaluated.

## 2.  Methods

### 2.1    Site description and instrumentation

The measurement was conducted at the campus of Peking University (39° 99' N, 116° 30' E). The
site is situated northeast of the Beijing city center and near two traffic roads, which represents a
typical urban and polluted area with fresh, anthropogenic emissions (Wang et al., 2017a). The
measurements were made on a building roof with a height of 20 m above the ground. Measurements
of VOC concentrations were performed using an automated gas chromatograph equipped with mass
spectrometry or flame ionization detectors (GC-MS/FID). There are 56 kinds of VOC are measured
in total, in which monoterpenes are not valid. NO$_x$ and O$_3$ were monitored by chemiluminescence
(Thermo Scientific, 42i-TLE) and UV photometric methods (Thermo Scientific, 49i), respectively. A
Tapered Element Oscillating Microbalance analyzer (TianHong, TH-2000Z1) was used to measure
the mass concentration of PM$_{2.5}$. The quality assurance and quality controls of data were
implemented regularly (Chen et al., 2020). Photolysis frequencies were obtained by the Tropospheric
Ultraviolet and Visible (TUV) model simulation. Hourly data were processed and used in the
following analysis.

### 2.2 Estimation of monoterpenes

Since the measurement data did not include monoterpenes (MNTs), we therefore use the measured
isoprene and modelled concentration ratio of monoterpene to isoprene in the same region of
measurement site (named as Factor, Eq. 2) to estimate the ambient monoterpene concentrations (Eq.
3). The Factor was obtained by the regional model (WRF/CMAQ), more details of the model
simulation setup can be found in Mao et al. (2022). We used the Factor to estimate monoterpenes
level rather than modelled monoterpene concentrations is due to the modelled isoprene is
systematically higher than that of observation (Fig. S1), thus the using of the modelled Factor may be
more reasonable. In Beijing, α-pinene and β-pinene were reported to have the highest abundance
among monoterpenes (Cheng et al., 2018), with higher emissions in summer (Wang et al., 2018b;
Xia and Xiao, 2019). Therefore, we approximated the averaged value of α-pinene and β-pinene
reaction rate coefficients with NO$_3$ in the following calculations. Since the emissions of
sesquiterpenes in BVOC are much lower than that of isoprene, monoterpenes and other BVOC, we
didn't consider NO$_3$ reactivity towards sesquiterpenes.
$$Factor = \frac{[MNT_{sim}]}{[ISO_{sim}]} \qquad \text{Eq. 2}$$
$$[MNT_{obs}] = [ISO_{obs}] \times Factor \qquad \text{Eq. 3}$$



**2.3 VOC oxidation rate by $NO_3$**
To study the reaction of $NO_3$ and VOC during the nighttime, we estimated the $NO_3$ concentrations by
steady-state calculation. This method is widely used to estimate the concentrations of short-lived
substances like $NO_3$, assuming its production and loss rates are balanced in a specific time range.
Given sufficient time, the steady state can be reached for $NO_3$ at night in which the production and
loss terms are approximately balanced (Brown, 2003; Crowley et al., 2010). The production terms of
$NO_3$ is the reaction of $NO_2$ and $O_3$, and the loss terms of $NO_3$ includes reactions with VOC, reaction
with NO, heterogeneous reaction, and photolysis. The steady-state $NO_3$ mixing ratios are expressed
by Eq. 4 (Brown and Stutz, 2012).
$$[NO_3]_{ss} = \frac{k_{NO_2+O_3}[NO_2][O_3]}{\sum k_i \times [VOC_i] + k_{NO+NO_3}[NO] + J_{NO_3} + k_{het}K_{eq}[NO_2]} \qquad \text{Eq. 4}$$
Where $J_{NO3}$ is the sum of the photolysis coefficients of the two photolysis reactions of $NO_3$. The $k_{het}$
is the heterogeneous uptake rate of $N_2O_5$ on the aerosol surface, which can be calculated by Eq. 5.
$$k_{het} = 0.25 \times \gamma \times S_a \times c \qquad \text{Eq. 5}$$
Where $\gamma$ is the dimensionless uptake coefficient of $N_2O_5$ parameterized by Eq. 6 (Evans and Jacob,
2005; Hallquist et al., 2003; Kane et al., 2001), Sa ($m^2$ $m^{-3}$) is the aerosol surface area density
estimated by the level of $PM_{2.5}$ (Wang et al., 2021), and $c$ is the mean molecular velocity of $N_2O_5$.
$$\gamma = \alpha \times 10^\beta$$
$$\alpha = 2.79 \times 10^{-4} + 1.3 \times 10^{-4} \times RH - 3.43 \times 10^{-6} \times RH^2 + 7.52 \times 10^{-8} \times RH^3$$
$$\beta = 4 \times 10^{-2} \times (T - 294) \ (T > 282K)$$
$$\beta = -0.48 \ (T < 282K) \qquad \text{Eq. 6}$$
The reaction rate coefficients of $NO_2$ and $O_3$, NO and $NO_3$, and the equilibrium constant for the
forward and reverse Reactions (R4) and (R5) are temperature dependent. We have adopted JPL
evaluation reports for the reaction rate coefficients. The time series of hourly related parameters in
estimating the steady-state $NO_3$ and the diurnal cycle of $NO_3$ concentrations were shown in Fig. S2
and Fig. S3. To compare the oxidation of $NO_3$ towards VOC with other oxidants, we estimated OH
concentrations by the slope that extracted from the measured OH and $J_{O1D}$ ($s^{-1}$) in North China (Tan
et al., 2017)) (Eq. 7), where $J_{O^1D}$ was obtained by the TUV model simulations. The VOC oxidation
rate and the ratio of VOC oxidized by $NO_3$ to the total oxidation rate can be calculated by Eq. 8.
$$[OH] = 4.1 \times 10^{11} cm^{-3} s^{-1} \times J_{O^1D} \qquad \text{Eq. 7}$$
$$R_{NO_3} \approx \frac{\sum k_i \times [VOC_i][NO_3]}{\sum k_i \times [VOC_i][OH] + \sum k_i \times [VOC_i][NO_3] + \sum k_i \times [VOC_i][O_3]} \qquad \text{Eq. 8}$$
where $k_i$ represents the corresponding reaction rate coefficients of different VOC with oxidants.



## 3. Results and discussion

### 3.1 NO₃ reactivity calculated by measured VOC

During the campaign, the hourly $k_{NO3}$ towards measured VOC (named as $k_{NO3\_mea}$) highly varied from <$10^{-4}$ to 0.083 s$^{-1}$ with campaign-averaged value (± standard deviation) of 0.0032 ± 0.0042 s$^{-1}$. The $k_{NO3\_mea}$ displayed a strong diel variation on annual average (Fig. S4). In previous studies, the NO₃ reactivity towards VOC was reported to be 0.024 ± 0.030 s$^{-1}$ on average in a suburban site in summer in North China (Yang et al., 2020); and highly varied between 0.005 - 0.3 s$^{-1}$ in mountaintop site in summer (Liebmann et al., 2018c). Our result is one order of magnitude lower, which may reflect the huge difference of $k_{NO3\_mea}$ in different environment and sampling time. Certainly, it may be attributed to the calculated $k_{NO3}$ here did not include some species, such as monoterpenes. The diurnal variations of $k_{NO3\_mea}$ had strong seasonal variability (Fig. S5). The diurnal variations in winter and spring were relatively weak, and the variations in summer and autumn were large, with clear peaks at 9:00-10:00 and 15:00, respectively. The $k_{NO3\_mea}$ in spring, summer and autumn reached the daily maximum value between 8:00 a.m. and 10:00 a.m. (spring: 0.0034 s$^{-1}$, summer: 0.0083 s$^{-1}$, autumn: 0.0057 s$^{-1}$). In winter, it reached the maximum value of 0.0033 s$^{-1}$ at about 22:00.

As shown in Fig. 1a, the $k_{NO3\_mea}$ reached the highest in August and lowest in February, which was largely affected by the level of isoprene and styrene. For example, isoprene contributed ~80% to the reactivity in August. The $k_{NO3\_mea}$ towards isoprene reached the maximum in August and the minimum in February, which was consistent with the previous reported change of isoprene concentrations in Beijing (Cheng et al., 2018). Figure 1b shows a large seasonal difference in $k_{NO3\_mea}$ with the average value of 0.0024 ± 0.0026 s$^{-1}$, 0.0067 ± 0.0066 s$^{-1}$, 0.0042 ± 0.0037 s$^{-1}$, 0.0027 ± 0.0028 s$^{-1}$ from spring to winter. Table S2 shows the specific contributions of top six species to $k_{NO3\_mea}$ in different seasons (and Fig. S4). Isoprene was the dominant species, accounting for 40.0%, 77.2% and 43.2% in spring, summer, and autumn. By comparison, styrene only played a leading role in winter, accounting for 39.8%. Of the species which contributed to $k_{NO3\_mea}$ in Beijing, isoprene and styrene contributed most to the overall $k_{NO3\_mea}$ (60%~90%) followed by cis-2-butene, trans-2-butene, trans-2-pentene and proplyene (5%~15%) with another individual VOC less than 2%. Our results are consistent with previous studies in Beijing that $k_{NO3}$ was mainly contributed by isoprene (Yang et al., 2020), indicating that the critical role of isoprene in NO₃ reactivity in Beijing. From summer to autumn, the dominant species changed from isoprene to styrene, while from winter to spring, the dominant species changed from styrene to isoprene. This indicated the AVOC and BVOC controls $k_{NO3\_mea}$ alternately. Overall, the $k_{NO3\_mea}$ displayed a characteristic of high in summer and autumn and low in winter and spring.



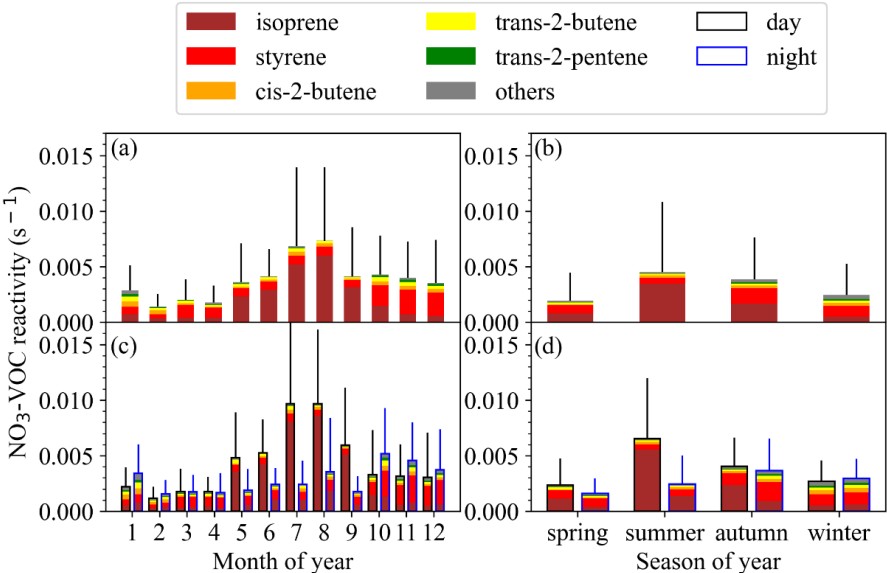

**Figure 1.** (a-b) Histograms of monthly and seasonal-averaged $k_{NO3\_mea}$ and the compositions. (c-d) Histograms of monthly and seasonal-averaged $k_{NO3\_mea}$ and the compositions divided into daytime (black frames) and nighttime (blue frames). The color denotes the contributions of different VOC species. The black and blue lines represent the error bars of the reactivity (± standard deviations).

Figure 1c-d showed the $k_{NO3\_mea}$ towards measured VOC display clear day-night differences in summer and winter, especially in summer. The $NO_3$ reactivity towards VOC in the daytime reached the value of 0.010 s$^{-1}$ in July and August, which was much higher than 0.002 s$^{-1}$ in the nighttime. The variations were mainly caused by the diel variations of isoprene concentrations. Reversely, the reactivity was higher at night and lower in the daytime in winter, which was due to the high AVOC level in the morning and at night (Lee and Wang, 2006). Specifically, styrene concentrations at night increased significantly in the stable nocturnal boundary layer, resulting in relatively higher reactivity.

In urban areas of Beijing, isoprene origins from anthropogenic and biological sources, in which the anthropogenic sources of isoprene are mainly traffic emissions (Li et al., 2013; Riba et al., 1987; Zou et al., 2015). In summer, isoprene mainly origins from plant, and in winter origins from the combustion of engine fuel. In spring and autumn, there are mixed effects of anthropogenic and biological origins (Li et al., 2013). The isoprene emissions of biological sources in Beijing were one order of magnitude larger than that of anthropogenic sources (Yuan et al., 2009). This indicates the concentrations of isoprene at the environmental level in the urban areas of Beijing is not affected by the traffic vehicles, but mainly by plants in Beijing (Cheng et al., 2018). As an aromatic hydrocarbon, styrene origins from both anthropogenic and biogenic sources in the atmosphere (Miller et al., 1994; Mogel et al., 2011; Schaeffer et al., 1996; Tang et al., 2000; Zielinska et al., 1996; Zilli et al., 2001), such as the laminar flame of engine fuel (Meng et al., 2016), industrial production (Radica et al., 2021) and other human activities. The dominant source of styrene in Beijing is the local vehicles emissions (Li et al., 2014). Some vegetation, such as evergreen and oleander, can release natural



styrene (Wu et al., 2014), however, due to the dense industrial distribution in the urban area and the
much lower level of these biogenic styrene compared with isoprene, we believed that the styrene in
the atmosphere in Beijing is mainly resulted from anthropogenic origins. It is believed that human
activities in winter, such as heating, gasoline and diesel combustion increased, meanwhile, the
reduction of temperature and radiation resulted in the reduction of biogenic isoprene emissions,
explained the conversion of dominance of $NO_3$ reactivity from summer to winter.
**3.2    Parameterization of $NO_3$ reactivity**
We examined the correlation of key VOC concentrations and $k_{NO3\_mea}$. Figure S6 gives the case in
January for example. To a certain extent, the variations of $k_{NO3\_mea}$ were closely linked to the
variations of the concentrations of main contributors. It is worth noting that in January,
trans-2-butene had a higher correlation coefficient with $k_{NO3\_mea}$, which exceeded that of isoprene and
styrene. This indicates that higher contributions may not imply stronger correlation. Fig. 2 shows the
correlation coefficients and the fitting equations between VOC concentrations and $k_{NO3}$ in each
month (detailed in Table S3). According to the correlation coefficients, we can select the strongest
indicator corresponding to the certain month as the variable of the parameterization method. Here we
didn't import the VOC with small contributions into the parameterization method, because these
indicators had no practical significance for $k_{NO3\_mea}$. In this way, we established the first
parameterization method by using the strongest indicator in each month and can be found in Table S3
(Eq. 10):
$$NO_3 \ reactivity_{sim1} = a_i \times [VOC_i] + b_i \qquad \text{Eq. 10}$$
where, $a_i$, $b_i$ and $[VOC_i]$ respectively represent the slope, the intercept and the VOC species
concentrations (ppbv) used for parameterization in each month. Throughout the year, the correlation
coefficients between isoprene concentrations and $k_{NO3\_mea}$ were high, ranging from 0.67 to 0.98,
especially in summer. The correlation coefficients between styrene concentrations and the reactivity
reached a maximum in autumn and winter, which can clearly display the indication of these two
species (isoprene and styrene) in different seasons.
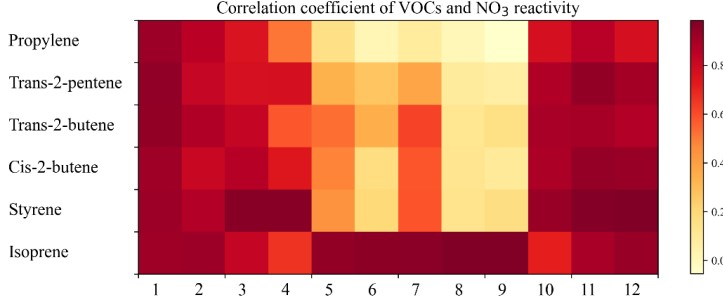

**Figure 2.** The thermodynamic diagram of the correlation between VOC concentrations and $k_{NO3\_mea}$.
Colored blocks indicate different correlations, by which the best indicator can be selected for
parameterization method of each month.



Besides the indicator parameterization method, we can also select only a part of VOC that contribute
most of $k_{NO3\_mea}$ as a representative. Here we approximated NO₃ reactivity towards total VOC to the
reactivity towards these top 6 species, namely isoprene, styrene, cis-2-butene, trans-2-butene,
trans-2-pentene and proplyene. Thus, the second parameterization method can be expressed by Eq.
273  11:

$$NO_3 \ reactivity_{sim2} = \sum_{i=1}^{6} k_i \times [VOC_i] \qquad \text{Eq. 11}$$
where $[VOC_i]$ is the VOC concentrations and $k_i$ is the corresponding reaction rate coefficients with
NO₃. It should be noted that this parameterization method of NO₃ reactivity towards VOC may be
localized.
To evaluate the effectiveness of the two parameterization methods established above, we estimated
the $k_{NO3}$ in the different time scale, and compared them with the determined $k_{NO3\_mea}$ by all measured
VOC. As shown in Fig. S7, both two methods can well capture the level and variations of $k_{NO3\_mea}$,
indicating the parametrization feasibility. Method 1 can easily and quickly estimate NO₃ reactivity
towards VOC by using a single indicator. In areas where NO₃ reactivity towards VOC is dominated
by a single VOC specie for a long time, such as forest areas, suburbs, rural areas (BVOC dominant),
this method would have a good performance. Method 2 had a better performance while more VOC
species are needed. In urban areas, especially in urban areas where the contributors had different
chemo diversity with strong seasonality, this method should be more suitable. Since the two methods
lower the bar for estimating NO₃ reactivity by using VOC measurement data, we can look into the
level of NO₃ reactivity by using the reported VOC measurement data in the past.
We collected the historical measurement data of VOC concentrations in Beijing (Supporting file. S1)
and estimated NO₃ reactivity by the parameterization methods. We found the level of NO₃ reactivity
mainly ranged from 0.001 to 0.1 s⁻¹ in Beijing in the past decades (Fig. 3). Due to the limitation of
data, we cannot find a trend of NO₃ reactivity before 2011. While during 2011-2020, large amount of
VOC data in urban Beijing presented and be collected in this study. We calculated the $k_{NO3\_mea}$ by
detailed VOC with respect to the data provided by the literatures, and estimated the NO₃ reactivity by
parameterization methods if the reported data in the literatures is limited. As shown in Fig. 3, an
overall decrease trend of NO₃ reactivity can be found during 2011-2020. We inferred that the level of
isoprene during this period may be varied small, since the biogenic emission unlikely to change
much. Thus, we proposed that the decrease of NO₃ reactivity during the past decade may be
attributed to the anthropogenic emission reduction of anthropogenic VOC. It should be noted that
this estimation suffers from the uncertainty, nevertheless, this trend and characterization of NO₃
reactivity in Beijing is helpful to understand the nighttime chemistry in Beijing.





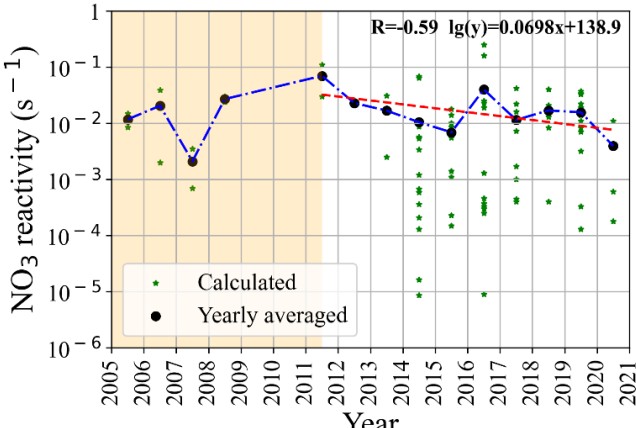


**Figure 3.** The reconstructed $NO_3$ reactivity calculated by the reported VOC concentrations in Beijing.
The averaged $NO_3$ reactivity calculated by the reported VOC data in each campaign plotted as star.
The yearly averaged $NO_3$ reactivity (black dot) between 2011-2019 shows a decline. It should be
noted that the monoterpenes are not considered here.

**3.3    $NO_3$ reactivity towards monoterpenes**

After taking MNTs into account, the total $k_{NO3}$ (named as $k_{NO3\_total}$) was greatly enlarged, with
campaign-averaged value of $0.0061 \pm 0.0088$ s$^{-1}$, resulting in our results comparable with previous
research results. The $NO_3$ reactivity towards MNTs (named as $k_{NO3\_MNTs}$) was higher in autumn and
winter and lower in spring and summer (Fig. S8). Considering the corresponding reactivity towards
monoterpenes, the total $NO_3$ reactivity towards VOC changed from (summer > autumn > winter >
spring) to (autumn > winter > summer > spring), highlights the impact of the monoterpene variations
on the reactivity. The $NO_3$ reactivity towards MNTs displayed significant differences between
daytime and nighttime (Fig. S8c-d). The reactivity at night in all months was higher than that in the
daytime, especially from October to January, highlights the role of biogenic monoterpenes in
nocturnal $NO_3$ chemistry (Li et al., 2013; Riba et al., 1987). To compare the measured and the total
$NO_3$ reactivity towards VOC, we calculated the fraction ($F_{MNTs}$) by Eq. 12.

$$F_{MNTs} = \frac{kNO3\_MNTs}{kNO3\_total} \qquad \text{Eq. 12}$$

Figure 4a displays the differences between the $k_{NO3\_mea}$ and $k_{NO3\_total}$. Monoterpenes were very
important for $NO_3$ reactivity, and the $F_{MNTs}$ varied from 40% to 80%, with strong seasonal variations.
The MNTs accounted for $NO_3$ reactivity nearly 80% in winter and spring. In the seasons when
isoprene no longer dominated, the measured reactivity accounted for a small fraction, and the
corresponding reactivity towards AVOC such as styrene was smaller than that of monoterpenes. As
shown in Fig. 4b, the measured VOC had high fractions in the daytime and low at night. Especially
in May and August. The measured VOC in the daytime accounted for more than 60% of $k_{NO3\_total}$,
which was closely related to the increasing concentrations of isoprene in the summer daytime. The




reactivity towards MNTs accounted for a large fraction of reactivity at night.

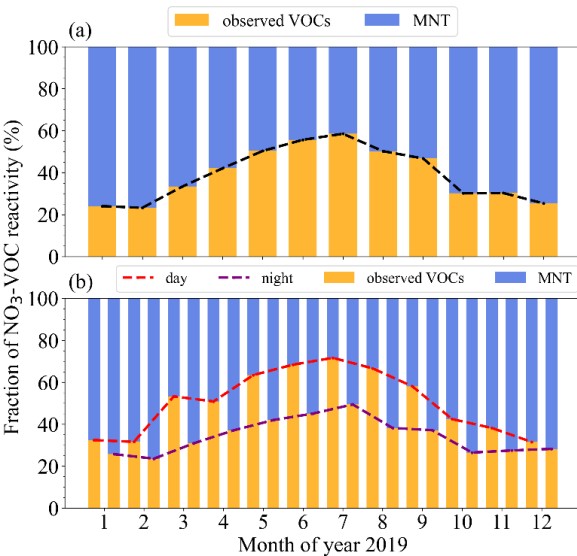


**Figure 4.** (a) Fractions of the $k_{NO3\_total}$. (b) Fractions of the $k_{NO3\_total}$ divided into daytime (left) and
nighttime (right). The colors on the stacked bar plot indicate the different fractions as they are
donated in the legend. The lines represent the monthly-averaged variations of the $NO_3$ reactivity
towards MNTs.
We updated the parameterization method established before by using the relationship between
reactivity and VOC concentrations including monoterpenes. The updated parameterization Method 1
used the same principle as introduced in Sect 3.2, with fitting slopes changing significantly (Fig. S9).
Table S4 gives the specific correlation coefficients between six key VOC concentrations and $k_{NO3\_total}$.
The updated Method 2 considered the sum contributions of six VOC and the estimated MNTs by
isoprene concentration. We reevaluated the two updated parameterization methods (single VOC and
six VOC, respectively). Overall, the performance of two methods are reasonable and the updated
Method 1 is better than that of Method 2 in general (Fig. S10).
**3.4    Nighttime VOC oxidation**
Here we examined the role of $NO_3$ in the VOC oxidation in Beijing 2019. As shown in Fig. 5, OH
oxidized most of VOC during the daytime, with the oxidation rate reached the maximum value of 0.6
pptv s$^{-1}$ in the afternoon. Compared with OH, the VOC oxidation rates by $O_3$ and $NO_3$ in the daytime
were remarkably lower. From 18:00 to 6:00, the characteristics of nocturnal chemical in Beijing were
significant. The ratios of VOC oxidized by $NO_3$ kept above 80%, the contribution of $O_3$ was
relatively weak, which is consistent with that reported in high NOx regions(Chen et al., 2019;
Edwards et al., 2017; Wang et al., 2018a). The VOC oxidation rate by $NO_3$ presented a single peak at
19:00 with the value of 0.25 pptv s$^{-1}$, which is the same magnitude as that by OH in the daytime,
illustrating the importance of $NO_3$ in VOC oxidation as shown in the previous studies (Wang et al.,



2017a), highlight the importance of nocturnal chemistry for organic nitrate and SOA formation.

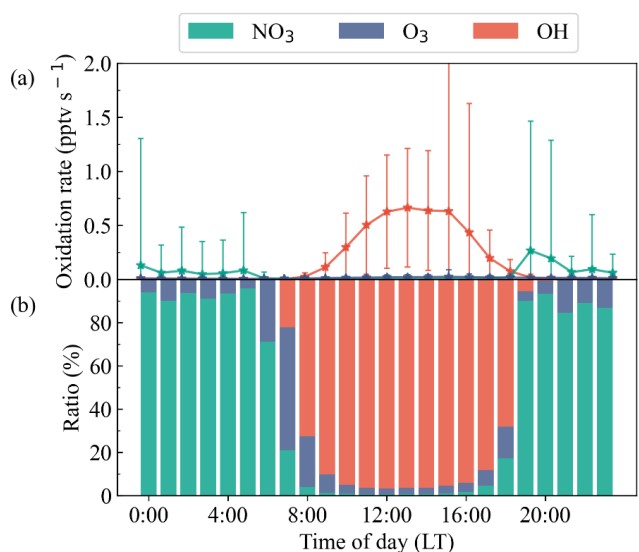

**Figure 5.** (a) Median diurnal profile of VOC oxidation rate by OH, NO₃ and O₃. The colored lines
are error bars (+standard deviation). (b) Fractions of VOC oxidation rate by atmospheric oxidants.

The VOC oxidation rate by $NO_3$ and oxidation fractions had strong seasonal variabilities in Beijing.
As shown in Fig. S11, the nighttime oxidation rate (summer > spring > autumn > winter) was
affected by $NO_3$ concentrations and the total $NO_3$ reactivity towards VOC. In summer, the $NO_3$
oxidation rate presented a single peak, with a maximum value of 0.7 pptv s$^{-1}$ at 20:00, and remained
around 0.1 pptv s$^{-1}$ at the rest of the night. The rate at 21:00-5:00 was relatively constant. The rate in
winter was lower, with the two maximum values of 0.06 pptv s$^{-1}$ presented at 19:00 and 4:00, which
were further lower than the average value of other three seasons. The results were good agreement
with the previous studies, in which the VOC oxidation rate by $NO_3$ concentrations contained high
from 19:00-23:00 (Wang et al., 2017b). There was a competition between $NO_3$ and $O_3$ in the
nighttime VOC oxidation in Beijing. Although the $NO_3$ oxidation rate at night was higher than that
of $O_3$ throughout the year, the changes of $O_3$ oxidation rate had a significant impact on the ratios of
VOC oxidized by $NO_3$. The ratios of nighttime VOC oxidized by $NO_3$ in Beijing were higher in
autumn, and then in spring, summer and winter. Although the $O_3$ concentrations in winter decreased,
the competitiveness of $NO_3$ in VOC oxidation decreased more due to the decline of $NO_3$
concentrations. The competitiveness of $O_3$ in VOC oxidation was relatively enhanced, resulting in a
significant decline in the ratios of VOC oxidized by $NO_3$.

**3.5 Regulation of nighttime VOC oxidation**

To understand the importance of nighttime VOC oxidized by $NO_3$, we explored the relationship
between the nocturnal oxidation ratios of $NO_3$ ($R_{NO3}$) and the nighttime concentrations of NO, $O_3$
and $NO_X$. It is found that a strong nonlinear relationship between them (shown in Figure 6). The



$R_{NO3}$ had negative correlation coefficients with NO concentrations. With the increase of NO
concentrations at night, the ratios decreased exponentially. When the NO concentrations increased at
low NO condition, it could cause a significant decline in the ratios of VOC oxidized by $NO_3$. While
at high NO condition, the ratios were not sensitive to the increase of NO concentrations (Fig. S12),
indicating that the nighttime NO concentrations in Beijing strongly controlled the ratios effectively. It
can be expected since the increase of NO concentrations controlled the $NO_3$ loss term, then caused
the decrease of $NO_3$ concentration. When the NO concentrations exceeded a threshold value, the $NO_3$
loss was totally dominated by NO.

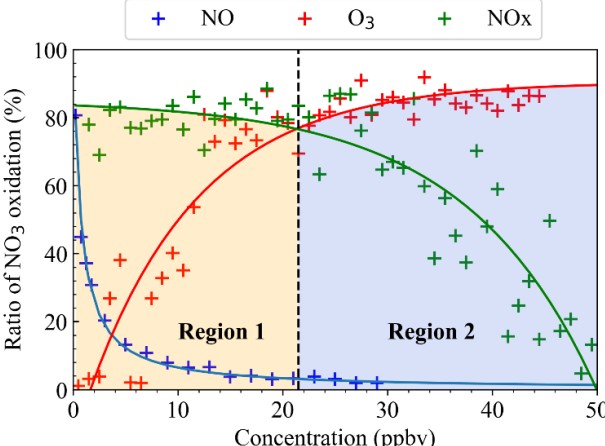


**Figure 6**. Fitting diagrams between the ratios of nighttime VOC oxidized by $NO_3$ and the
concentrations of NO, $O_3$ and $NO_X$. In Region 1, the ratio is more sensitive to $O_3$, while less sensitive
to $NO_X$. In Region 2, it is more sensitive to $NO_X$, while less sensitive to $O_3$.

The ratios of nighttime VOC oxidized by $NO_3$ also had a strong nonlinear relationship with $O_3$ and
NOx concentrations. $O_3$ concentrations have one positive and one negative contribution to the $R_{NO3}$.
The positive effect is that increasing $O_3$ concentration increase the $NO_3$ production rate, which
increase the $NO_3$ steady-state concentrations then increase the ratios. And the negative is increasing
$O_3$ concentrations increase the reaction rate between VOC and $O_3$, which increase the
competitiveness of $O_3$ in VOC oxidation then decrease the ratios. Figure. S12 also shows the
relationship between the $R_{NO3}$ and the concentrations of $O_3$. While $O_3$ concentrations below 21.5
ppbv, the ratios were very sensitive to $O_3$ level, which fast increased with $O_3$ concentrations. While
the ratio become not sensitive and remained relatively constant when the $O_3$ concentrations exceeded
21.5 ppbv. It can be explained that when the $O_3$ concentrations were low, the $NO_3$ production rate was
more sensitive to the increase of $O_3$ concentrations. In this case, $O_3$ mainly affects the ratios
positively. When the $O_3$ concentrations were high, the positive effect of $O_3$ tended to be constant,
indicates the two opposite effects overall keep in balance.
When the $NO_X$ concentrations were low (e.g., <21.5 ppbv), the $R_{NO3}$ were less sensitive to $NO_X$,
remaining relatively constant with the further increase. It is believed that the increase of $NO_3$ loss



rates through the $N_2O_5$ heterogeneous reaction and the NO reaction were kept in balance with the
$NO_3$ production rate increased by $NO_2$ concentrations. At high NOx condition, the ratios sensitively
decreased with the increase of $NO_X$ concentrations, which is explained that the increase of $NO_3$ loss
rates by NO, resulting in a decline in the ratios.
To better understand the nonlinear effect of $NO_2$ and $O_3$ on the nighttime VOC oxidation, we further
explored the effect of $O_3$ concentration on the ratios existing in different concentrations of $NO_2$. As
shown in Fig. S13, in higher concentrations of $NO_2$, the threshold of lower $O_3$ concentrations were
required for the $R_{NO3}$ to become constant, which reflected the couple influence of $NO_2$ and $O_3$ on
nighttime VOC oxidation through the nonlinear response, and indicated that in the environment
richen in $NO_2$, nocturnal $NO_3$ chemistry easily tended to be more dominant.
**4.    Conclusions and implications**
In this study, we showed the $NO_3$ reactivity towards measured VOC highly varied with strong
seasonal differences, which was mainly driven by isoprene concentrations. The top 6 contributors to
the measured $NO_3$ reactivity towards VOC were isoprene, styrene, cis-2-butene, trans-2-butene,
trans-2-pentene and propylene. Among them, isoprene and styrene contributed most of the reactivity.
In addition, monoterpenes are proposed to be a significant source of $NO_3$ reactivity. Recently studies
showed the anthropogenic emissions contributes significantly to the ambient MNTs concentrations
by biomass burning, traffic and volatile chemical product emissions in the urban regions, it would
further enhance the importance of nocturnal $NO_3$ oxidation (Coggon et al., 2021; Nelson et al., 2021;
Peng et al., 2022; Qin et al., 2020; Wang et al., 2022). It should be noted that the estimated
contributions of MNTs only considered the biogenic emissions and may be represent the lower bias,
thus we highlight the importance of field observation of MNTs for advancing the understanding the
nighttime $NO_3$ chemistry. In addition, it should be noted that we didn't take the contributions of
OVOC into account, since the reaction rate coefficients of OVOC with $NO_3$ are small (Ambrose et
al., 2007).
Looking insight to the trend and evolution of detailed $NO_3$ chemistry is very scare, but it can really
helpful to understand response of the nocturnal chemistry on the emission change at a large time
scale. Limited by the non-extensive and non-continuous observation, we cannot obtain the long-time
measurement of all the VOC species in multiple sites. Since isoprene and styrene are good indicators
of $NO_3$ reactivity in different seasons, at least as we shown in urban Beijing, those can be used to
estimate the $NO_3$ reactivity towards VOC to reestablish the long-term trend of $NO_3$ reactivity in
urban regions for further evaluation of its history of nighttime chemistry. We admitted that the
estimation of $NO_3$ reactivity trend may be highly uncertain, but this attempt may be very helpful to
know the level and overall change of nighttime chemistry.
We showed that $NO_3$ dominated the nighttime VOC oxidation in Beijing, but the oxidation ratio had
a strong nonlinear relationship with $O_3$ and $NO_X$ concentrations. With the $NO_2$ concentrations
decrease, the threshold values of $O_3$ between sensitive regime and non-sensitive regime tended to
increase, indicative of the nighttime oxidation by $NO_3$ would be more easily affected by the level of
$O_3$ with the implement of sustaining NOx reduction in the future. The threshold values of $O_3$ can
provide an effective basis for the measures to control nocturnal chemical and secondary organic



aerosols pollution in the typical urban region.
**Code/Data availability.** The datasets used in this study are available from the corresponding author
upon request (wanghch27@mail.sysu.edu.cn; k.lu@pku.edu.cn).
**Author contributions.** H.C.W. and K.D.L. designed the study. H.J.H. and H.C.W. analyzed the data
with input from J.W., Z.L.Z., X.Z.X., T.Y.Z., X.R.C., X.L., M.M.Q. provided the modelled
monoterpene and isoprene data, X.L., L.M.Z., M.H., and Y.H.Z. organized this field campaign and
provided the field measurement dataset. H.J.H. and H.C.W. wrote the paper with input from K.D.L.
**Competing interests**. The authors declare that they have no conflicts of interest.
**Acknowledgments**. This project is supported by the National Natural Science Foundation of China
(42175111, 21976006). Thanks for the data contributed by field campaign team.

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
