# Peer review of "Variation and Trend of Nitrate radical reactivity towards 1"

_EGUsphere, 2023_

## Author Comment (AC1)

Response to Editors and Reviewers

We appreciate the reviewers for their careful reading and constructive comments on our manuscript. As detailed below, the reviewer's comments are shown in black, our response to the comments is in blue. New or modified text is in red.

All the line numbers refer to Manuscript ID: acp-2023-622.

**Referee 1**

The author conducted a one-year VOCs observation in Beijing, attempting to illustrate NO3 chemistry, especially the oxidation of VOCs by NO3. Then, a parameterization method was established to attempt to construct the long-term reactivity of NO3 using VOCs data. It was found that since 2011, the NO3 reactivity of VOCs in Beijing has been decreasing year by year, and there is a significant correlation with VOCs emission reduction. The overall research objectives of the article are clear, the research methods are appropriate, and the research conclusions are relatively reliable. However, there are still issues that need to be clarified, as follows.

Thanks for the review's constructive comments.

Major comments:

1.  Correlation diagram, meaning of horizontal and vertical coordinates. For example, Figure S1. Why use simulated values as the abscissa? Generally, the abscissa is the reference value, or the true value. In this case, it is obvious that observation values should be used as the abscissa. Please review the entire article by the author.

    We have reviewed the entire article and revised all figures containing simulated and actual values. In Figure S1 (Figure S2 now), we used the observed concentration of isoprene as the abscissa and the concentration of isoprene simulated by WRF/CMAQ as the ordinate to more clearly demonstrate the relationship between the simulated and observed values. Since the reactivity containing monoterpenes was estimated, in Figure S8 and Figure S10 of section 2.2, we used the reactivity as the ordinate and used VOC concentration as the abscissa, which can better demonstrate the relationship between VOC concentration and NO$_3$ reactivity towards VOCs.

    In brief, we revised the figures as follows (Figure S2 for example).

[Figure]

**Figure S2**. The intercomparing of measured and simulated isoprene in year 2019 in urban Beijing, which shows the overall overestimation of the modelling result compared with the observation.

2. Why is temperature and humidity linear between April and May? Is there no observed data? If not, please delete it. The uptake coefficient of N2O5 also has this issue.

Yes. We have removed the "linear" section because there were no values in the observation data of temperature and relative humidity from April to May. Due to the fitting relationship between the uptake coefficient with temperature and relative humidity in the heterogeneous reaction of $N_2O_5$, the uptake coefficient of $N_2O_5$ had no values from April to May, so the steady-state concentration of $NO_3$ also had no values.

We have revised Figure S2 (now Figure S3) as follows

[Figure]

3. The red lines of d and e in Figure S2 are both NO3 losses, with the former being the total loss and the latter being the loss in reaction with VOCs. It seems that there is a significant difference between the two orders of magnitude, is it due to the participation of heterogeneous processes?

It isn't due to the participation of heterogeneous processes. In our study, we found that the concentration of NO in Beijing were relatively high, which led to $NO_3$ mainly lost through the reaction with NO (which can be clearly seen in the fitting diagram of $R_{NO3}$ and NO concentration in Section 3.5 (Section 3.4 now)). Therefore, the significant difference between these two orders of magnitude is due to the higher concentrations of NO which dominated the loss of $NO_3$. We added the explanation of the loss of $NO_3$ in the caption as follows.

Figure S3. (a) Time series of concentrations of NO, $NO_2$ and $O_3$. (b) Time series of thermodynamic temperature and relative humidity. (c) Time series of the aerosol surface area densities and dimensionless uptake coefficients of $N_2O_5$, (d) Time series of $NO_3$ production rate and $NO_3$ loss term (the reaction with NO dominated the reactivity). (e) Time series of $NO_3$ stationary-state concentrations and the $NO_3$ loss term through the reaction with VOC (actual $NO_3$ reactivity towards VOC). All the data was averaged with the time resolution of 1 day.

4. Figure S7 shows that current parameterization methods can capture changes in monthly and daily averages, but cannot capture changes in hourly averages. Why? After adding terpenes in Figure S10, the fitting results actually look worse. Why?

Thanks for the interesting comment. The parameterization scheme selected the indicator species on a monthly scale, and the derivation of parameterization used the hourly data set. Therefore, the fitting result should be best in the monthly and daily scale since the goal of the parameterization is reproduced the result on a monthly trend. We confirmed that the estimated averaged diel variations of reactivity cannot capture the changes in the highest time resolution with hourly averages. However, due to the relatively stable indicating effect of VOC every month, the estimated averaged diel variations of reactivity based on months are also consistent with the actual averaged level and magnitude.

After adding the contribution of monoterpenes, the errors from the simulated concentration of monoterpenes were introduced in the actual reactivity, and there were also certain errors in parameterized estimation. The two types of errors together caused larger uncertainty in the fitting results in Figure S10 (Figure S11 now), but the averaged level and magnitude of the fitting results are relatively consistent. In future studies, the observation data of monoterpene will help to further update and improve the performance of these parameterization schemes.

5. The article discusses the NO3 reactivity of VOCs, but there is no VOCs concentration sequence diagram in the article, especially the proportion of different components, seasonal changes, daily changes, etc. In addition, the author needs to explain the detection method of VOCs and what are the pre freezing stages? Detect which species and so on, for example, indicate 56 PAMS species.

Thanks for the comments. We depicted the averaged-diel profiles, averaged-seasonal profiles and time series of concentrations of alkanes, alkenes, aromatics, and isoprene, styrene, cis-2-butene, trans-2-butene, and trans-2-pentene, and added them in the supporting information (as Figure S1 in the revised version). We added an explanation of detection method of VOC and pre-freezing stages, and classified and numbered the 56 observed VOCs in Table S1.

Line 124. The volatile organic compounds were pretreated by pre-freezing and collected in the deactivated quartz empty capillary at extreme-low temperature (-150 ℃), then heated and

delivered into the analysis system. After separation by the double chromatographic column, the low carbon compounds C2-C4 were detected by the FID detector, and the high carbon compounds C5-C10 were detected by the MS detector. There are 56 kinds of VOC are measured in total and accounted in this study (listed in Table S1 and the concentrations are depicted in Figure S1), in which monoterpenes measurement are not valid.
Please find the Table S1 in the updated SI.

[Figure]

**Figure S1.** The yearly-averaged diel profiles (a), monthly-averaged profiles (c) and time series (e) of concentrations of alkanes, alkenes and aromatics. The yearly-averaged diel profiles (b), monthly-averaged profiles (d) and time series (f) of concentrations of isoprene, styrene, cis-2-butene, trans-2-butene and trans-2-pentene. The colors indicate the different VOC as they are donated in the legend.

6. The article emphasizes that the simulation results of terpenes come from WRF-CMAQ. What specific simulation scheme is used? Given the important contribution of terpenes, it is recommended that the author provide a detailed explanation.

We agreed with this suggestion and added more description about the model simulation setup in the revised manuscript as follows.

Line143. The regional model CMAQ (Community Multiscale Air Quality) version 5.2 was applied to simulate air quality in the eastern China, with a horizontal resolution of 36 km. Specifically, the gas-phase mechanism of SAPRC-07 and aerosol module AERO6 were used. The meteorological fields were provided by Weather Research & Forecasting (WRF) Model version 4.2. The biogenic emissions were simulated by the MEGANv2.1, which was driven by WRF as well, and the emissions of open burning were estimated with FINN. The MEIC

emission inventory for 2019 (obtained via private communication) was used to represent anthropogenic emissions over China, while the emissions in the areas outside China were provided by the REAS v3.2 inventory simulation.

7. The author used the ratio of isoprene to terpene to calculate the concentration of terpene, which may require supporting evidence. Firstly, the emission of terpenes from biological sources is related to temperature, while the emission of isoprene from biological sources is related to temperature and radiation. There may be a relationship between the two at night, but the relationship may not be significant during the day. Secondly, the author also emphasizes that motor vehicles emit a large amount of isoprene, and if the ratio is used to determine terpenes, it may lead to an overestimation of terpenes.

We highly appreciate the suggestion. As you mentioned, the factors affecting the concentrations of isoprene and monoterpene are different, so we did not use the same parameters to simulate the concentrations of the two. Instead, we used the model to simulate the relationship (or the ratio) between the two to reflect this systematic difference. To be honest, we propose that this method is reasonable and feasible while there is no more evidence since long term measurement of both isoprene and monoterpene is very scarce. This is the best what we can do to estimate the contribution of terpene. We hope to provide more evidence to prove this relationship in future research.

The isoprene emissions from motor vehicles are important parts of anthropogenic isoprene emissions in Beijing, but the anthropogenic isoprene emissions can be ignored compared to biological sources. Many studies have shown that anthropogenic isoprene emissions are less important in Beijing: The isoprene emissions of biological sources in Beijing were one order of magnetic larger than that of anthropogenic sources (Yuan et al., 2009). This indicates the concentration of isoprene at the environmental level in the urban areas of Beijing is not affected by the traffic vehicles, but mainly by plants in Beijing (Cheng et al., 2018). Therefore, the anthropogenic isoprene emissions can be ignored and the concentrations obtained by model were able to estimate without significant overestimation in our study.

Otherwise, since the previous introduction of isoprene emissions from motor vehicles in the article may provide readers with a confused understanding of isoprene in Beijing, we have deleted it in the paragraph.

8. Since WRF-CMAQ can simulate OH concentration, why use J1D to calculate OH concentration?

Here the parameterization based on the $JO^1D$ data is based on the relationship derived from the field observation. Actually, we cannot say which OH dataset would be better to reflect the real OH level in the atmosphere. Therefore, we compared the WRF-CMAQ modelled OH

and the JO$^1$D parameterized results. As shown in the following figure, the inter-comparison confirmed that these two methods had a similar and consistent performance with a controllable difference (although the model simulation is systemically higher than fitted result with 41%). We tested the usage of modelled OH in the calculation as shown in the following analysis, although the CMAQ modelled OH enlarged the daytime oxidation, it doesn't have a significant impact on the results of VOC nocturnal oxidation. Therefore, we did not change the use of JO$^1$D parameterization in the revised manuscript.

[Figure]

9. Isn't Equation 11 the method for calculating NO3 reactivity? What is the difference between this equation and equation 1? It seems that the difference is only slightly fewer species.

   Yes. Parameterization scheme 2 is equivalent to calculate the addition of NO$_3$ reactivity towards important contributing VOC. For the Beijing region, it is the addition of NO$_3$ reactivity towards isoprene, styrene, cis-2-butene, trans-2-butene, trans-2-pentene and propylene. Since not all observed VOCs were calculated, it is called the second type of parameterization method. The significance of this method is to greatly reduce the requirements for VOC observation, which ignores VOC species with small contributions to obtain NO$_3$ reactivity towards VOC.

10. Since terpenes have made significant contributions, I think it is meaningless without adding the explanation of terpenes in sections 3.1 and 3.2. I suggest deleting section 3.3 and merging it into 3.1 and 3.2 for discussion and explanation. The author can use the simulation results of long-term terpenes to illustrate the interannual trend of NO3 reactivity of VOCs in Beijing.

    We added the monoterpene reactivity and its parameterization scheme in section 3.3 to the corresponding part of sections 3.1 and 3.2, respectively.

    The simulation results of long-term terpenes are lacking, but we speculate that the changes of long-term concentrations of BVOC such as terpenes may not be significant as you suggested that it is close related to the temperature, therefore the interannual change of NO$_3$ reactivity towards VOC may not be clearly demonstrated using long-term modeled terpene concentrations.

We adjusted section 3.3 according to your comments as follows.

Line 257 (section 3.1)

The $NO_3$ reactivity towards MNTs (named as $k_{NO3\_MNTs}$) was estimated by the method mentioned in section 2.2. After taking MNTs into account, the total $k_{NO3}$ (named as $k_{NO3\_total}$) was greatly enlarged, with campaign-averaged value of $0.0061 \pm 0.0088$ s$^{-1}$, resulting in our results comparable with previous research results. The $NO_3$ reactivity towards MNTs was higher in autumn and winter and lower in spring and summer (Fig. S7). Considering the corresponding reactivity towards monoterpenes, the total $NO_3$ reactivity towards VOC changed from (summer > autumn > winter > spring) to (autumn > winter > summer > spring), highlighting the impact of the monoterpene variations on the reactivity. The $NO_3$ reactivity towards MNTs displayed significant differences between daytime and nighttime (Fig. S7c-d). The reactivity at night in all months was higher than that in the daytime, especially from October to January, highlights the role of biogenic monoterpenes in nocturnal $NO_3$ chemistry (Li et al., 2013; Riba et al., 1987). To evaluate the contribution of monoterpenes to the total $k_{NO3}$, we calculated the fraction (FMNTs) by Eq. 12.

$$F_{MNTs} = \frac{k_{NO_3\_MNTs}}{k_{NO_3\_total}} \qquad \text{Eq. 12}$$

Figure 2a displays the differences between the $k_{NO3\_mea}$ and $k_{NO3\_total}$. Monoterpenes were very important for $NO_3$ reactivity, and the $F_{MNTs}$ varied from 40% to 80%, with strong seasonal variations. The MNTs accounted for $NO_3$ reactivity nearly 80% in winter and spring. In the seasons when isoprene no longer dominated, the measured reactivity accounted for a small fraction, and the corresponding reactivity towards AVOC such as styrene was smaller than that of monoterpenes. As shown in Fig. 2b, the measured VOC had high fractions in the daytime and low at night. Especially in May and August. The measured VOC in the daytime accounted for more than 60% of $k_{NO3\_total}$, which was closely related to the increasing concentrations of isoprene in the summer daytime. The reactivity towards MNTs accounted for a large fraction of reactivity at night.

[Figure]

Figure 2. (a) Fractions of the $k_{NO3\_total}$. (b) Fractions of the $k_{NO3\_total}$ divided into daytime (left) and nighttime (right). The colors on the stacked bar plot indicate the different fractions as they are donated in the legend. The lines represent the monthly-averaged variations of the $NO_3$ reactivity towards MNTs.

Line 346 (Section 3.2)

After taking MNTs into account, we updated the parameterization method established before by using the relationship between reactivity and VOC concentrations including monoterpenes. The updated parameterization Method 1 used the same principle as introduced in Sect 3.2, with fitting slopes changing significantly (Figure. S10). Table S5 gives the specific correlation coefficients between six key VOC concentrations and $k_{NO3\_total}$. The updated Method 2 considered the sum contributions of six VOC and the estimated MNTs by isoprene concentration. We reevaluated the two updated parameterization methods (single VOC and six VOC, respectively). Overall, the performance of two methods are reasonable and the updated Method 1 is better than that of Method 2 in general (Fig. S11). For robustness, we evaluated this parameterization on datasets of other years (shown in Fig S12).

---

## Author Comment (AC2)

**Response to Editors and Reviewers**

We appreciate the reviewers for their careful reading and constructive comments on our manuscript. As detailed below, the reviewer's comments are shown in black, our response to the comments is in blue. New or modified text is in red.

All the line numbers refer to Manuscript ID: acp-2023-622.

**Referee 2**

The chemistry of NO3 with VOC affects the budget of nocturnal SOA, and regulates regional photo chemistry indirectly. This study presents a detailed analysis about the nitrate radical reactivity towards VOC based on the one-year VOC measurement in a typical urban site. The level, compositions and seasonal variation of NO3 reactivity are well characterized. The results showed isoprene and styrene dominated NO3 reactivity on average, and proposed a month-resolved parameterization scheme to predict NO3 reactivity by using one or several VOCs species data. They tried to rebuilt the dataset of NO3 reactivity by using the scheme and collecting the historical VOC measurement data, and showed an overall decrease trend in recent years. Although this result may be highly uncertain, it provided a new and interesting perspective of the nighttime chemistry in a long-term period. This topic certainly within the scope of ACP and the manuscript is overall well-written. I would like to recommend it be accepted after the authors address the following comments listed below:

**Thanks for the review's overall positive comments.**

Major comments:

 In section 3.2, the authors constructed a parameterization to estimate NO3-VOCs reactivity using one or several VOCs concentration based on one-year data in 2019. For robustness, it is better to evaluate this parameterization on datasets of other years, since you have collected the historical data of VOC concentrations in Beijing for several years before 2019. We highly acknowledge this suggestion. We used the collected historical data of VOC concentrations in Beijing for several years before 2019 and selected isoprene, styrene and

other indicators to estimate the corresponding  $NO_3$  reactivity towards VOC by parameterization method 1. Error analysis was conducted and it was found that the parameterization method 1 estimated the reactivity with relatively small error, indicating that our parameterization schemes based on the indicators concentrations in Beijing is reasonable.

In brief, we have added histograms to demonstrate the effectiveness of parameterization method 1 as follows.

**Figure S12.** Histograms of actual NO3 reactivity towards VOC and reactivity estimated through parameterization method 1 at different times in Beijing. The figure displays the indicators introduced in parameterization method 1 and the relative errors of estimation at different times. The (up/down) arrows represent the estimated effect (overestimation/underestimation) of the parameterized method 1.

Line 354. We evaluated this parameterization on datasets of other years (shown in Fig S12) and showed a robustness performance.

2. 5: The title 'Regulation of nighttime VOC oxidation' is somewhat vague for readers to follow. Maybe it is better to use 'The relationship between NO/Ox and nighttime VOC oxidation by NO3 radical'. I also found that the dependence of RNO3 on NOx has shown similar messages with the dependence on NO, which could be confusing in this figure as it is not related to the regions defined. Since the RNO3 has a relatively good exponential correlation with NO, I am wondering if Fig 6 can first focus on NO dependence and divide it into three regions, which might be regarded as NO-limited, transition and NO-saturated. In NO-saturated region, RNO3 is closed to zero and shows no dependence on both NO2 and O3. In the other two regions, RNO3 can reach up to 80%, and then it could be of interest to look into how the attribution of Ox influence the RNO3 variation.

Thanks. We have adjusted the previous title of Section 3.5 (Section 3.4 now) to "Relationship between  $NOx/O_3$  and nocturnal VOC oxidation by  $NO_3$  radical ".

We divided the nonlinear region of NO into two regions: NO-limited and NO-saturated region (NO-transited region was merged into the NO-limited region). Within the NO-limited region (NO concentration < 7 ppbv), we fitted the oxidation ratio with O3 and NOx concentrations. It was found that within the NO-limited region, the oxidation ratio was sensitive to O3 at 0-25 ppbv and NOx at 25-50 ppbv. Finally, we displayed three regions: O3- limited region within NO-limited region, NOx-limited region within NO-limited region and NO-saturated region. There is no significant difference between the modified fitting results and the previous one, indicating good consistency in this result. In brief, we have revised Figure 6 as follows to

better demonstrate the synergistic control relationship and process of different species on the oxidation ratios.

**Figure 6.** Fitting diagrams between the ratios of nighttime VOC oxidized by  $NO_3$  and the concentrations of NO (a),  $O_3$  (b) and  $NO_X$  (c). The light pink scattered dots represent the oxidation ratios at different concentrations and the solid dots represent the median value of each bin of oxidation ratios corresponding to each concentration range. Colored dot lines represent fitting results of the solid median dots. And the black dot line in each panel shows a threshold to divide the curve into two regimes. In (a), the regime divided into NO-limited (<7 ppbv) and NO-saturated (>7 ppbv), in (b) and (c), a threshold of 25 ppbv divide the curves into  $NO_X$  (O3) limited and saturated regimes. The results showed in (b) and (c) are representative of low NO condition (<7 ppbv).

Technical corrections:

- 1. Line 25: Please change 'the' to 'that'
- 2. Corrected accordingly.
- Line 49: This equilibrium reaction can also take place during the day. We revised the wording throughout the manuscript accordingly.
- Line 62: Please delete 'type' Deleted accordingly.
- Line 86: Please change ';' to '.'. Corrected accordingly.
- Line 102: Please change 'varies' a verb to an adjective. Corrected accordingly.
- Line 117: Suggest adding 'newly' before 'proposed'. We added 'newly' accordingly.
- Line 118: The word 'regulation' is a bit of vague here.
   We rewrote it as follows

Line 115. At last, the nocturnal VOC oxidation by NO3 during different seasons was further

evaluated.

 Line 138~139: Does this factor vary temporally? If so, please provide the specific values Yes. We added a table to show the temporal variation of this factor as follows.

|       | Jan  | Feb  | Mar  | Apr  | May  | Jun  | Jul  | Aug  | Sep  | Oct  | Nov  | Dec  |
|-------|------|------|------|------|------|------|------|------|------|------|------|------|
| 0:00  | 1.66 | 1.66 | 1.56 | 1.26 | 1.56 | 1.34 | 1.34 | 1.49 | 1.20 | 1.68 | 1.66 | 1.84 |
| 1:00  | 2.05 | 1.96 | 2.01 | 1.69 | 2.20 | 1.71 | 1.69 | 2.00 | 1.46 | 2.05 | 1.93 | 2.16 |
| 2:00  | 2.21 | 2.11 | 2.22 | 1.94 | 2.59 | 1.95 | 1.99 | 2.51 | 1.75 | 2.24 | 2.08 | 2.28 |
| 3:00  | 2.26 | 2.15 | 2.30 | 2.04 | 2.72 | 2.06 | 2.26 | 2.91 | 1.96 | 2.34 | 2.18 | 2.31 |
| 4:00  | 2.66 | 2.52 | 2.86 | 2.48 | 3.19 | 2.06 | 2.68 | 3.45 | 2.37 | 2.85 | 2.63 | 2.69 |
| 5:00  | 3.72 | 3.57 | 4.45 | 3.54 | 1.24 | 0.68 | 1.00 | 2.07 | 3.28 | 4.38 | 3.71 | 3.79 |
| 6:00  | 3.69 | 3.67 | 4.46 | 3.56 | 0.81 | 0.49 | 0.71 | 1.39 | 3.34 | 4.53 | 3.76 | 3.92 |
| 7:00  | 2.61 | 2.66 | 2.69 | 2.04 | 0.45 | 0.31 | 0.32 | 0.43 | 0.73 | 2.16 | 2.60 | 2.81 |
| 8:00  | 1.73 | 1.76 | 1.39 | 0.94 | 0.18 | 0.13 | 0.14 | 0.16 | 0.24 | 0.98 | 1.64 | 1.85 |
| 9:00  | 1.36 | 1.35 | 0.95 | 0.62 | 0.13 | 0.09 | 0.10 | 0.10 | 0.14 | 0.63 | 1.26 | 1.42 |
| 10:00 | 1.20 | 1.18 | 0.80 | 0.49 | 0.11 | 0.08 | 0.08 | 0.09 | 0.11 | 0.49 | 1.07 | 1.24 |
| 11:00 | 1.24 | 1.23 | 0.84 | 0.50 | 0.11 | 0.08 | 0.07 | 0.08 | 0.10 | 0.46 | 1.05 | 1.29 |
| 12:00 | 1.17 | 1.18 | 0.79 | 0.46 | 0.11 | 0.07 | 0.06 | 0.08 | 0.09 | 0.42 | 0.97 | 1.22 |
| 13:00 | 1.01 | 1.04 | 0.68 | 0.39 | 0.10 | 0.07 | 0.06 | 0.07 | 0.09 | 0.36 | 0.86 | 1.08 |
| 14:00 | 0.90 | 0.93 | 0.60 | 0.35 | 0.09 | 0.06 | 0.06 | 0.07 | 0.08 | 0.32 | 0.78 | 0.98 |
| 15:00 | 0.93 | 0.94 | 0.62 | 0.37 | 0.10 | 0.07 | 0.07 | 0.07 | 0.09 | 0.34 | 0.82 | 1.03 |
| 16:00 | 1.32 | 1.26 | 0.86 | 0.52 | 0.13 | 0.08 | 0.08 | 0.08 | 0.11 | 0.54 | 1.20 | 1.46 |
| 17:00 | 1.86 | 1.80 | 1.20 | 0.74 | 0.17 | 0.09 | 0.09 | 0.11 | 0.17 | 0.97 | 1.83 | 2.04 |
| 18:00 | 2.10 | 2.19 | 1.47 | 0.87 | 0.21 | 0.11 | 0.11 | 0.14 | 0.29 | 1.41 | 2.14 | 2.28 |
| 19:00 | 2.37 | 2.43 | 1.97 | 1.17 | 0.34 | 0.14 | 0.14 | 0.23 | 0.58 | 1.91 | 2.42 | 2.58 |
| 20:00 | 2.45 | 2.46 | 2.16 | 1.49 | 0.74 | 0.44 | 0.34 | 0.50 | 0.87 | 2.20 | 2.48 | 2.69 |
| 21:00 | 2.13 | 2.16 | 1.82 | 1.33 | 0.94 | 0.84 | 0.77 | 0.77 | 0.94 | 2.03 | 2.16 | 2.38 |
| 22:00 | 1.59 | 1.70 | 1.32 | 0.95 | 0.88 | 0.88 | 0.95 | 0.91 | 0.85 | 1.59 | 1.70 | 1.85 |
| 23:00 | 1.47 | 1.57 | 1.32 | 1.01 | 1.13 | 1.08 | 1.15 | 1.16 | 0.98 | 1.49 | 1.59 | 1.72 |

**Table S2.** The averaged diurnal variations of Factor  $(=\frac{MNT_{sim}}{ISO_{sim}})$  during different months.

10. Line 140: Please delete the first 'is'.

Corrected accordingly.

11. Line 190: Please add the 'reason that' before 'the calculated'.

Revised accordingly.

- 12. Line 205: Please change 'of' to 'among'.We polished the language throughout the manuscript accordingly.
- Line 207: What is the 'another individual VOC'? I guess it means other? Yes. We revised it to 'other VOC'.
- Line 213~214: I think the conclusion sentence here is not so necessary here. Or the authors could just put it at the beginning of this paragraph. Thanks, we deleted it accordingly.

Overall, the  $k_{NO3\_mea}$  displayed a characteristic of high in summer and autumn and low in winter and spring

15. Fig 1: In my opinion, the information in this figure is a little bit overlapped by presenting NO3-VOC reactivity levels in both months and seasons of a year. I think the months of year style is enough to demonstrate the temporal variations of reactivity. And the seasonal variations can be just provided in the text. In addition, it is not easy to identify daytime and nighttime reactivity with thin frames in black and blue. Shadow padding could be better.

We appreciate for the reviewer's suggestion to revise Figure S1 in order to clearly display the daily and diurnal average levels of NO3 reactivity towards VOC in different seasons. The revised figure is as follows.

---

## Author Response (AR2)

Editor:The content is OK, but the quality of figures should be improved (such as type, shadow in Fig. 5 et al. ) before publication.

Response: Thanks, we improved all the six figures accordingly, and unified MNT to MNTs throughout the manuscript. In addition, we adjusted the order of funding